# Visual Privacy Auditing with Diffusion Models

**Kristian Schwethelm**                                                                 *k.schwethelm@tum.de*
*Chair for AI in Healthcare and Medicine, Technical University of Munich (TUM) and TUM University Hospital*

**Johannes Kaiser**                                                                 *johannes.kaiser@tum.de*
*Chair for AI in Healthcare and Medicine, Technical University of Munich (TUM) and TUM University Hospital*

**Moritz Knolle**                                                                 *moritz.knolle@tum.de*
*Chair for AI in Healthcare and Medicine, Technical University of Munich (TUM) and TUM University Hospital*

**Sarah Lockfisch**                                                                 *sarah.lockfisch@tum.de*
*Chair for AI in Healthcare and Medicine, Technical University of Munich (TUM) and TUM University Hospital*

**Daniel Rückert**                                                                 *daniel.rueckert@tum.de*
*Chair for AI in Healthcare and Medicine, Technical University of Munich (TUM) and TUM University Hospital*
*Department of Computing, Imperial College London, UK*
*Munich Center for Machine Learning (MCML), Munich, Germany*

**Alexander Ziller**                                                                 *alex.ziller@tum.de*
*Chair for AI in Healthcare and Medicine, Technical University of Munich (TUM) and TUM University Hospital*

**Reviewed on OpenReview:** *https://openreview.net/forum?id=D3DA7pgpvn*

## Abstract

Data reconstruction attacks on machine learning models pose a substantial threat to privacy, potentially leaking sensitive information. Although defending against such attacks using differential privacy (DP) provides theoretical guarantees, determining appropriate DP parameters remains challenging. Current formal guarantees on the success of data reconstruction suffer from overly stringent assumptions regarding adversary knowledge about the target data, particularly in the image domain, raising questions about their real-world applicability. In this work, we empirically investigate this discrepancy by introducing a reconstruction attack based on diffusion models (DMs) that only assumes adversary access to real-world image priors and specifically targets the DP defense. We find that (1) real-world data priors significantly influence reconstruction success, (2) current reconstruction bounds do not model the risk posed by data priors well, and (3) DMs can serve as heuristic auditing tools for visualizing privacy leakage.

## 1 Introduction

The widespread collection of sensitive data—including personal identities, private locations, and medical conditions—has raised critical privacy concerns, particularly in machine learning (ML) where models can leak private information from their training data. While differential privacy (DP) (Dwork & Roth, 2014) has emerged as the gold standard for providing formal privacy guarantees, the practical effectiveness of these guarantees against real-world privacy attacks remains uncertain, especially in data reconstruction scenarios where adversaries attempt to recover complete data records. This uncertainty poses a substantial challenge for practitioners facing privacy-utility trade-offs, as stronger privacy protections typically reduce model performance. To deploy DP techniques effectively, practitioners need a clear understanding of how the mathematical privacy guarantees translate to practical protection of sensitive information.

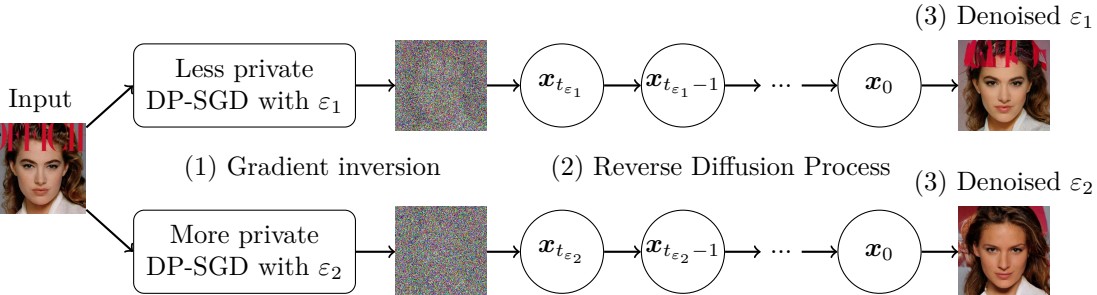

Figure 1: (1) Our reconstruction attack first extracts a noisy image from a DP algorithm with privacy guarantee $\varepsilon_n$ using, *e.g.*, gradient inversion on DP-SGD. (2) Then, it employs a DM for reconstruction by initiating its reverse diffusion process from a specific intermediate state $\boldsymbol{x}_{t_{\varepsilon_n}}$. (3) We demonstrate DMs' strong utility for reconstruction and visual auditing, aiding communication with non-experts. In this example, it is possible to infer that $\varepsilon_1$ offers little privacy protection, allowing accurate reconstruction, while $\varepsilon_2$ safeguards certain details but still allows disclosure of high-level personal attributes.

Attempts to address this issue have led to the development of formal upper bounds on data reconstruction success under DP, aiming to enable practitioners to assess the effect of DP guarantees on the maximum fidelity achievable by an adversary's reconstruction attack. Central to such bounds are the formalized threat models, which define the capabilities of potential adversaries and the scenarios under which the bounds hold. Due to the complexity of mathematically formalizing practical scenarios, overly pessimistic threat models that account for the most powerful (worst-case) attacks have been adopted (Balle et al., 2022; Guo et al., 2022). Although these bounds offer generality and hold against all possible attacks, they potentially overestimate the threat in real-world scenarios (Ziller et al., 2024a). For instance, while Hayes et al. (2023) demonstrate the near-tightness of reconstruction robustness (ReRo) bounds (Balle et al., 2022), their analysis assumes a *highly informed* adversary with access to a prior that includes the complete target record—a scenario that is unlikely to occur in practice.

As a result, there is an ongoing effort to formulate practical reconstruction bounds tailored to realistic attack scenarios. Ziller et al. (2024b) derive formal bounds on error measures for a specific reconstruction attack on DP-SGD (Song et al., 2013; Abadi et al., 2016) training, assuming an *uninformed* adversary capable of selecting model architecture and hyperparameters but lacking prior knowledge about the data. However, this threat model may be overly optimistic in domains where data priors are common, particularly in the image domain, where the underlying structure and characteristics of the data are well understood. This raises critical questions: *How do realistic data priors influence the effectiveness of data reconstruction attacks against differentially private machine learning, and can existing reconstruction bounds capture these threats?*

In this work, we empirically investigate the effectiveness of DP in defending against image reconstruction attacks that leverage *real-world data priors* (*i.e.*, domain-specific knowledge about the underlying data distribution). We compare our findings with the theoretical guarantees given by the (worst-case) ReRo bound (Hayes et al., 2023) and Ziller et al. (2024b). For our study, we extend a well-established gradient attack method that involves adversarial modification of the model architecture—an approach proven effective in, *e.g.*, federated learning (Fowl et al., 2022; Boenisch et al., 2023a) and backdooring pretrained models (Feng & Tramèr, 2024)—and incorporate image priors approximating the underlying data distribution of the reconstruction target.

Our attack extension builds upon a principal finding: *under DP-SGD, this gradient attack is equivalent to reconstructing an image perturbed with additive noise* (Boenisch et al., 2023b; Ziller et al., 2024b). We exploit this characteristic by leveraging strong image priors learned by diffusion models (DMs) (Ho et al., 2020; Dhariwal & Nichol, 2021) to denoise the reconstructions (see Fig. 1). This enables us to specifically target DP-SGD's reliance on noise perturbations and model adversarial access to real-world data priors.

Our findings reveal the substantial influence of data priors on reconstruction success, with their impact varying depending on the strength of the prior (inherent distribution shift). We find that works on reconstruction

bounds do not model these observations well. Instead, they rely on simplified assumptions not reflective of real-world scenarios. Beyond highlighting these theoretical limitations, we propose a practical application of our method: a *visual auditing tool* that complements formal DP guarantees, making privacy risks tangible and interpretable for non-technical stakeholders. By bridging theoretical bounds and practical privacy evaluation, our work contributes to a better understanding of DP's real-world implications and highlights crucial directions for developing more realistic privacy guarantees.

Our main contributions can be summarized as follows:

1. We demonstrate that the efficacy of realistic reconstruction depends on the strength of the data prior, which is not adequately represented by current theoretical bounds. This highlights a significant gap between theory and practice in privacy guarantees.

2. We introduce an image reconstruction attack leveraging diffusion models to model adversaries with realistic data priors. Our results reveal the significant threat such priors pose in disclosing private information under DP-SGD.

3. We empirically identify privacy parameters necessary to defend against our attack, and demonstrate its efficacy as a heuristic tool for visually auditing privacy risks.

## 2 Background and Related Work

### 2.1 Differential Privacy Guarantees

Differential privacy (DP) (Dwork & Roth, 2014) is a formal guarantee that provably bounds privacy leakage from computations on datasets.

**Definition 1.** A randomized algorithm (mechanism) $\mathcal{M}$ satisfies $(\varepsilon, \delta)$-DP if, for any pairs of adjacent datasets $D \simeq D'$ that differ in a single sample and all sets of outcomes $\mathcal{S} \subseteq \mathrm{Range}(\mathcal{M})$, it holds that:

$$\Pr[\mathcal{M}(D) \in \mathcal{S}] \leq e^{\varepsilon} \Pr[\mathcal{M}(D') \in \mathcal{S}] + \delta. \tag{1}$$

Intuitively, DP limits the influence of an individual sample on the algorithm's outcome. In this work, we focus on the Gaussian mechanism (GM), a standard DP mechanism often used in machine learning (ML). GMs introduce controlled randomization to mask the contribution of a sample through the addition of i.i.d. Gaussian noise $\mathcal{N}(0, \Delta_2^2 \sigma^2 \mathbf{I})$, where the variance of the noise is calibrated to the privacy guarantee using noise multiplier $\sigma$ and the mechanism's global L$_2$-sensitivity $\Delta_2$ (Balle & Wang, 2018).

The standard DP threat model assumes an adversary with complete knowledge, except for the noise realization and whether $D$ or $D'$ was used to compute the mechanism's outcome. The latter aspect naturally aligns DP with membership inference attacks, which aim to determine whether a specific individual's data was part of the dataset (Yeom et al., 2018).

The prevailing approach to implementing DP in ML is DP-Stochastic Gradient Descent (DP-SGD) (Song et al., 2013; Abadi et al., 2016), which limits the privacy leakage from training. DP-SGD is a modified version of SGD that enforces an upper bound on sensitivity $\Delta_2 = C$ by clipping per-sample gradients to an upper norm bound $C$ and adding calibrated i.i.d. Gaussian noise $\mathcal{N}(0, C^2 \sigma^2 \mathbf{I})$.

**Interpreting DP Guarantees.** In DP, the level of privacy preservation provided by an algorithm is typically quantified using parameters such as $\varepsilon$ and $\delta$. However, a more operationally interpretable way of quantifying privacy is by attributing practical risk against certain attacks under specific threat models. Recent work has advanced our understanding of practical DP effects through auditing techniques (Lokna et al., 2023; Nasr et al., 2023; Steinke et al., 2023; 2024), relaxed threat models (Nasr et al., 2021; Kaissis et al., 2023b; Ziller et al., 2024a), novel privacy attacks (Geiping et al., 2020; Boenisch et al., 2023a; Feng & Tramèr, 2024), and deployment strategies (Ponomareva et al., 2023; Cummings et al., 2024). However, the impact of adversarial access to real-world data priors on privacy risk remains largely unexplored. Furthermore, while membership inference attacks have been extensively studied, the threats posed by reconstruction

attacks—which aim to recover complete data records—are less understood. Our work addresses these gaps by leveraging powerful data priors to evaluate reconstruction risks under DP, providing practitioners with empirical insights for navigating privacy-utility trade-offs.

## 2.2 Bounding Data Reconstruction Success

Data reconstruction attacks on ML models pose a critical privacy risk by attempting to recover complete data records. While DP mechanisms primarily target membership inference protection, they inherently also defend against broader privacy breaches, including data reconstruction. However, in scenarios where membership information is considered insensitive or even public knowledge, practitioners might consider relaxing DP guarantees to improve model utility. This has motivated several theoretical bounds on reconstruction success (Guo et al., 2022; Stock et al., 2022; Balle et al., 2022; Ziller et al., 2024b). Yet, these bounds may not fully capture the threat of practical attack scenarios, particularly when adversaries possess prior knowledge about the data. Our work complements these theoretical results by providing an empirical framework to assess reconstruction risks in practical settings.

**Reconstruction Robustness (ReRo).**  ReRo (Balle et al., 2022; Hayes et al., 2023; Kaissis et al., 2023a) provides a formal upper bound on the probability of a successful data reconstruction attack.

**Definition 2.** A randomized algorithm (mechanism) $\mathcal{M}$ satisfies $(\eta, \gamma)$-ReRo if, for any reconstruction attack $R$ on the algorithm's output $\omega$, any dataset $D_- \cup \{z\}$, where $z$ denotes the reconstruction target sampled from prior $\pi$, fixed error function $\rho$, and baseline success probability $\kappa_{\pi,\rho}(\eta)$, it holds that:

$$\kappa_{\pi,\rho}(\eta) \le \mathbb{P}_{z \sim \pi, \, \omega \sim \mathcal{M}(D_- \cup \{z\})}(\rho(z, R(\omega)) \le \eta) \le \gamma. \tag{2}$$

ReRo adopts the DP threat model with the slight modification that only a fixed part of the dataset $D_-$ is known to the adversary, while the added reconstruction target $z$ is not. However, the adversary has some prior knowledge $\pi$ about the target, which, informally, serves as a reference distribution for $z$.

Given the difficulty in determining $\rho$, $\eta$, and $\pi$, as well as approximating $\kappa_{\pi,\rho}(\eta)$, Hayes et al. (2023) introduced a worst-case ReRo definition ($(0, \gamma)$-ReRo) based on *sample matching*. Let $\rho = \mathbb{1}(z \ne R(\omega))$, $\eta = 0$, $\kappa_{\pi,\rho}(\eta) = 1/n$, and $\pi$ be a uniform distribution over a discrete set of $n$ candidate samples $\{z_{\text{target}}, z_1, \ldots, z_{n-1}\}$. Then, the adversary's task reduces to re-identifying the target by matching the observation $\omega$ to the correct sample from the prior set, which results in a simplified "reconstruction" setting. Despite its limitations, we use $(0, \gamma)$-ReRo as our theoretical baseline since it remains the only computationally viable implementation of ReRo. For clarity, we use $(0, \gamma)$-ReRo and ReRo interchangeably.

**Uninformed Data Reconstruction.**  Ziller et al. (2024b) introduced formal bounds on error metrics for a specific data reconstruction attack on DP-SGD training. They assume an uninformed adversary with no prior knowledge about the data but with the ability to observe gradient updates and modify the model architecture and training hyperparameters. By exploiting these capabilities, they showed that a worst-case adversary can replace the architecture to maximize privacy vulnerabilities. Specifically, they reduce the model to a single fully connected layer without bias, where the output of the layer directly represents the loss: $\ell = \mathbf{W}x$, with $\mathbf{W}$ denoting the weights and $x$ the input data. This architecture allows direct reconstruction of the input by inverting the observed gradients $x_{\text{rec}} = \frac{\partial \ell}{\partial \mathbf{W}} = x$. However, the application of clipping and additive Gaussian noise on the gradients introduced by DP-SGD perturbs the reconstruction, leading to:

$$x_{\text{rec}} = \frac{x}{\max(||x||_2/C, 1)} + \xi, \text{ with } \xi = \mathcal{N}(\mathbf{0}, C^2\sigma^2\mathbf{I}). \tag{3}$$

Leveraging the closed-form solution of this uninformed reconstruction attack, Ziller et al. formally analyze the reconstruction success and, *e.g.*, bound the expected mean squared error (MSE): $\text{MSE}(x, x_{\text{rec}}) \ge C^2\sigma^2$.

### 2.3 Diffusion Models (DMs)

Diffusion models, particularly the denoising diffusion probabilistic models (DDPMs) (Ho et al., 2020), have gained significant attention in recent years. DDPMs rely on a forward diffusion process that step-wise perturbs a signal (image) $\boldsymbol{x}_0 \sim q(\boldsymbol{x}_0)$ with additive i.i.d. Gaussian noise until the noise predominates. Mathematically, the forward process is described by:

$$\boldsymbol{x}_t = \sqrt{1 - \beta_t}\boldsymbol{x}_{t-1} + \sqrt{\beta_t}\boldsymbol{\epsilon}_{t-1}, \text{ with } \boldsymbol{\epsilon}_{t-1} \sim \mathcal{N}(\mathbf{0}, \mathbf{I}), \tag{4}$$

where the noise schedule $\beta_t \in (0, 1)$ controls both the variance of the noise and the factor reducing the signal at step $t = \{1, 2, \ldots, T\}$. By defining $\alpha_t := 1 - \beta_t$, $\bar{\alpha}_t := \prod_{s=1}^{t} \alpha_s$, and given the underlying Markov chain $q(\boldsymbol{x}_1, \ldots, \boldsymbol{x}_T \mid \boldsymbol{x}_0) = \prod_{t=1}^{T} q(\boldsymbol{x}_t \mid \boldsymbol{x}_{t-1})$, the noisy latent variables $\boldsymbol{x}_t$ can be conditioned on $\boldsymbol{x}_0$:

$$\boldsymbol{x}_t = \sqrt{\bar{\alpha}_t}\boldsymbol{x}_0 + \sqrt{1 - \bar{\alpha}_t}\boldsymbol{\epsilon}, \text{ with } \boldsymbol{\epsilon} \sim \mathcal{N}(\mathbf{0}, \mathbf{I}). \tag{5}$$

The reverse process, used for generating new signals, employs a neural network to approximate the intractable distribution $q(\boldsymbol{x}_{t-1} \mid \boldsymbol{x}_t)$ and predict the sampled noise $\boldsymbol{\epsilon}$. Given a large number of steps $T$ and well-behaved schedules of $\beta_t$, $\boldsymbol{x}_T$ converges to a standard Gaussian. Thus, a signal can be generated by initiating the reverse process with a standard Gaussian sample and iterative denoising.

**Image Denoising with Diffusion Models.** Generative image denoising strategies leveraging diffusion models have demonstrated state-of-the-art perceptual quality in natural (Xie et al., 2023; Pearl et al., 2023; Yang et al., 2023) and medical imaging (Xiang et al., 2023b;a; Chung et al., 2023). Notably, in the broader field of image restoration, diffusion models have also demonstrated efficacy in tasks like super-resolution, colorization, and inpainting (Li et al., 2023). In contrast to these works, we do not aim to enhance image quality by removing some minor natural noise. Instead, we aim to recover *private* information from deliberately perturbed images with *substantial* noise scales introduced to provide DP guarantees.

**Image Reconstruction with Diffusion Models.** Concurrent work (Huang et al., 2024; Liu et al., 2025) has also explored the use of DMs for maximizing attack success in image reconstruction. However, our study adopts a broader approach by investigating how data priors influence these attacks in the context of DP. We also analyze how such attacks align with or challenge existing theoretical reconstruction bounds, providing a deeper understanding of their implications for privacy guarantees.

## 3 Method

In this section, we present our methodology by formally introducing the problem and describing our approach to leveraging diffusion models (DMs) for image reconstruction.

### 3.1 Problem Definition

**Threat Model.** We study a common attack scenario on DP-SGD training where an adversary can manipulate the model architecture, hyperparameters, and observe training gradients to reconstruct private images. Beyond these capabilities, we assume the adversary has access to realistic image priors, *i.e.*, statistical knowledge about natural image features (such as textures, edges, and color gradients) or domain-specific patterns (like facial features or medical imaging characteristics).

**Base attack.** Our work builds on the analytical attack introduced by Fowl et al. (2022), which enables near-perfect data reconstruction from training gradients by placing a fully connected (imprint) layer at the model's front. For a fully connected layer ($\boldsymbol{y} = \mathbf{W}\boldsymbol{x} + \boldsymbol{b}$), where $\boldsymbol{x}$ is the input, $\boldsymbol{W}^i$ denotes a weight row, and $b^i$ a bias parameter, the gradients of the loss $\mathcal{L}$ with respect to a single row of weights and the bias are:

$$\nabla_{\boldsymbol{W}^i}\mathcal{L} = \frac{\partial \mathcal{L}}{\partial y^i}\frac{\partial y^i}{\partial \boldsymbol{W}^i} = \frac{\partial \mathcal{L}}{\partial y^i}\boldsymbol{x}, \quad \nabla_{b^i}\mathcal{L} = \frac{\partial \mathcal{L}}{\partial y^i}\frac{\partial y^i}{\partial b^i} = \frac{\partial \mathcal{L}}{\partial y^i}. \tag{6}$$

Element-wise division of these gradients ($\frac{\partial \mathcal{L}}{\partial y^i} \boldsymbol{x} \oslash \frac{\partial \mathcal{L}}{\partial y^i} = \boldsymbol{x}$) perfectly recovers the input, showing that gradients can directly encode the training data. When applied to DP-SGD, this attack yields a scaled, noisy version of the target image (see (Boenisch et al., 2023b; Ziller et al., 2024b)). While the original attack results in noise from a ratio distribution due to the division of Gaussian random variables, Ziller et al. (2024b) show how to modify the attack to achieve Gaussian noise instead.

**Adversarial Problem Statement.** We consider a scenario where an adversary extracts a perturbed image from a differentially private computation. In a standard DP-SGD setting, this extraction can be achieved by either maliciously setting a batch size of 1 to directly obtain per-sample gradients or by applying common techniques to extract per-sample gradients from accumulated gradients (Fowl et al., 2022; Boenisch et al., 2023a;b). The privatized observation is represented as:

$$\boldsymbol{x}_{\mathrm{priv}} = \frac{1}{\lambda}\boldsymbol{x} + \boldsymbol{\xi}, \tag{7}$$

where $\lambda = \max(\|\boldsymbol{x}\|_2/C, 1)$ denotes the clipping factor, $\boldsymbol{x} \sim q(\boldsymbol{x})$ denotes the original image sampled from data distribution $q(\boldsymbol{x})$, and $\boldsymbol{\xi}$ is sampled from i.i.d. Gaussian noise $\mathcal{N}(\boldsymbol{0}, C^2\sigma^2\mathbf{I})$. The adversary's goal is to reconstruct the private information in $\boldsymbol{x}$ from $\boldsymbol{x}_{\mathrm{priv}}$. Following the attack scenario described above, the reconstruction task reduces to a denoising problem. This formulation allows us to assess the practical privacy leakage and determine sufficient noise levels for protection.

## 3.2 Private Image Reconstruction with Diffusion Models

Diffusion models (DMs) learn powerful image priors that closely approximate complex data distributions by solving denoising tasks. Their ability to combine observed features with learned statistical patterns makes them highly effective at reconstructing corrupted information. Additionally, DMs can handle various noise levels without retraining, making them well-suited for our work, investigating the effectiveness of different noise perturbations. We leverage these strengths to develop a reconstruction attack that exploits DP-SGD's reliance on noise-based privacy mechanisms.

Given the inverse problem in Eq. (7), we define the posterior over the observation as $q(\boldsymbol{x} \mid \boldsymbol{x}_{\mathrm{priv}})$. We approximate this posterior using DMs and leverage their Markov chain to initiate the reverse process from a conditional intermediate state $p_\theta(\boldsymbol{x}_{t-1} \mid \boldsymbol{x}_{\mathrm{priv}})$ instead of pure noise until the original image is recovered, *i.e.*, $\boldsymbol{x}_0 \approx \boldsymbol{x}$. The easiest choice to integrate $\boldsymbol{x}_{\mathrm{priv}}$ into the reverse process is adopting the Variance Exploding (VE) form of DMs (Song et al., 2021b):

$$\boldsymbol{x}_t = \boldsymbol{x}_0 + \sigma_t \boldsymbol{\epsilon}, \text{ with } \boldsymbol{\epsilon} \sim \mathcal{N}(\boldsymbol{0}, \mathbf{I}), \tag{8}$$

with variance schedule[1] $\{\sigma_t^2\}_{t=1}^T$ and $\sigma_T^2 \to \infty$. Notice that this formulation does not reduce the signal by $\sqrt{\bar{\alpha}_t}$, which is also the case in Eq. (7). However, the VE form complicates hyperparameter tuning since standard DMs employ the Variance Preserving (VP) form, where $\sqrt{1 - \bar{\alpha}_T} \to 1$ (see Eq. (5)). To utilize the VP form, we use the equivalence between the two forms (Kawar et al., 2022) and define the starting point of the reverse process as follows:

$$\boldsymbol{x}_{t_{\mathrm{start}}} = \frac{1}{\sqrt{1 + \sigma_{t_{\mathrm{start}}}^2}}\boldsymbol{x}_{\mathrm{priv}} = \frac{1}{\sqrt{1 + \sigma_{t_{\mathrm{start}}}^2}}\left(\frac{1}{\lambda}\boldsymbol{x} + \boldsymbol{\xi}\right), \tag{9}$$

where $t_{\mathrm{start}}$ denotes the starting step in the DM's noise schedule.

**Handling the Clipping Factor $\lambda$.** An unknown parameter in Eq. (9) is the linear scalar $\lambda$ introduced by the clipping operation of DP-SGD. This parameter scales down the image, reducing its brightness and value range. In a realistic scenario, the exact value of $\lambda$ is unknown to the adversary. However, given that $\lambda$ represents a single value and images are typically characterized by a constrained range of color values, the

---

[1]Note that $\sigma_t$ denotes the noise variance relative to $\boldsymbol{x}_0$, which differs from $\beta_t$ and $\bar{\alpha}_t$ in the standard DM definition (see Sec. 2.3), as well as from $\sigma$ in DP (see Sec. 2.1).

adversary can easily approximate $\lambda$ through normalization or trial-and-error (see Appendix A). Therefore, we assume the worst-case scenario, wherein the adversary successfully recovers the exact value of $\lambda$.

We stress that knowing $\lambda$ comes with little advantage to the adversary, as it only enables them to rescale the image after perturbation, which increases the noise sample $\xi$ by factor $\lambda$. Thus, the signal-to-noise ratio remains unchanged. Combining our assumption with Eq. (9) yields:

$$\boldsymbol{x}_{t_{\text{start}}} = \frac{\lambda}{\sqrt{1 + \sigma_{t_{\text{start}}}^2}} \boldsymbol{x}_{\text{priv}} = \frac{1}{\sqrt{1 + \sigma_{t_{\text{start}}}^2}} (\boldsymbol{x} + \lambda\boldsymbol{\xi}). \tag{10}$$

**Markov Chain Matching.** The Markov chain of (discrete) DMs is based on a pre-defined noise schedule $\{\beta_t\}_{t=1}^T$ and, therefore, does not contain a state for all possible noise variances. Thus, to initiate the reverse process from a perturbed image with variance $\hat{\sigma}^2 = C^2\sigma^2\lambda^2$, we must compute the variance schedule

$$\sigma_t = \sqrt{\frac{1}{\prod_{s=1}^t (1 - \beta_t)} - 1} = \sqrt{\frac{1}{\bar{\alpha}_t} - 1} \tag{11}$$

and search the next largest state $t_{\text{start}}$ under the condition $\sigma_{t_{\text{start}}} > \hat{\sigma}$.

**Enforcing Data Consistency.** The stochastic generative process of DMs introduces randomness after each denoising step, increasing sample diversity. However, this is not desirable in reconstruction problems, where the results should closely resemble the original. Therefore, we enforce data consistency by adopting the deterministic generation process of denoising diffusion implicit models (DDIMs) (Song et al., 2021a), which has been shown to retain image features throughout the generation process:

$$\boldsymbol{x}_{t-1} = \sqrt{\bar{\alpha}_{t-1}} \left( \frac{\boldsymbol{x}_t - \sqrt{1 - \bar{\alpha}_t} \cdot \epsilon_\theta^{(t)}(\boldsymbol{x}_t)}{\sqrt{\bar{\alpha}_t}} \right) + \sqrt{1 - \bar{\alpha}_{t-1}} \cdot \epsilon_\theta^{(t)}(\boldsymbol{x}_t). \tag{12}$$

Data consistency can also be enforced by conditioning every step of the reverse process on $\boldsymbol{x}_{\text{priv}}$ by, *e.g.*, concatenating the low-quality sample to each latent state $\boldsymbol{x}_t$, yielding the posterior distribution $p_\theta(\boldsymbol{x}_{t-1} \mid \boldsymbol{x}_{\text{priv}}, \boldsymbol{x}_t)$. However, our scenario considers extreme cases where the images are heavily perturbed. Conditioning with such low-quality images causes harmful effects on the generation of DMs (Li et al., 2023). Thus, we forgo such an approach.

Algorithm 1 summarizes our method.

---

**Algorithm 1:** Private Image Reconstruction with DMs

---

**Require:** $\boldsymbol{x}_{\text{priv}} = 1/\lambda\boldsymbol{x} + \boldsymbol{\xi}$, with $\boldsymbol{\xi} \sim \mathcal{N}(0, C^2\sigma^2\mathbf{I})$, noise schedule $\bar{\alpha}_t$, model $\theta$

1: $\sigma_t = \sqrt{\frac{1}{\bar{\alpha}_t} - 1}$             ▷ Variance schedule

2: $\boldsymbol{x}'_{\text{priv}} = \lambda\boldsymbol{x}_{\text{priv}}$             ▷ Rescaling

3: $t_{\text{start}} = \arg\min_t(\sigma_t - C\sigma\lambda) \; \forall \; \sigma_t > C\sigma\lambda$             ▷ Markov chain matching

4: $\boldsymbol{x}_{t_{\text{start}}} = \frac{1}{\sqrt{1 + \sigma_{t_{\text{start}}}^2}} \bar{\boldsymbol{x}}_{\text{priv}}$             ▷ Reparameterization

5: **for** $t = t_{\text{start}}, \ldots, 1$ **do**             ▷ Step-wise denoising

6:      $\boldsymbol{x}_{t-1} = \sqrt{\bar{\alpha}_{t-1}} \left( \frac{\boldsymbol{x}_t - \sqrt{1 - \bar{\alpha}_t} \cdot \boldsymbol{\epsilon}_\theta^{(t)}(\boldsymbol{x}_t)}{\sqrt{\bar{\alpha}_t}} \right) + \sqrt{1 - \bar{\alpha}_{t-1}} \cdot \boldsymbol{\epsilon}_\theta^{(t)}(\boldsymbol{x}_t),$

7: **end for**

---

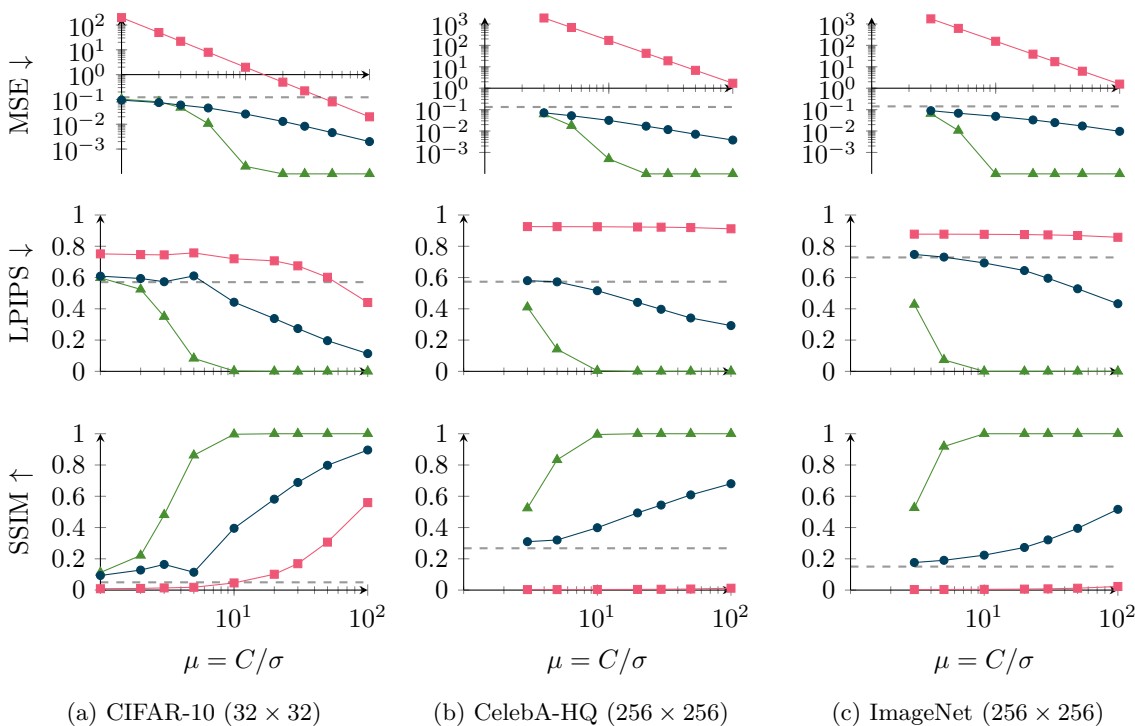

Figure 2: Average similarity of image reconstructions. We compute the similarity between the original and the reconstructed images from our DM attack (*blue* •), the attack of (Ziller et al., 2024b) (*red* ■), and the ReRo attack (Hayes et al., 2023) (*green* ▲). For $\mu < 3$, CelebA-HQ and ImageNet images exceed the maximal noise variance $\sigma_T$ in the schedule; thus, no results can be given. The *dashed* line represents average similarity between test images, indicating at which point reconstructions become unrelated to the original.

## 4 Experiments

This section compares the data reconstruction success of prevailing theoretical reconstruction bounds and our practical attack leveraging image priors learned by diffusion models (DMs). Additionally, it investigates the effectiveness of using DMs to specifically target the DP-SGD defense in scenarios with limited target access and weaker attack assumptions. For experimental details and ablation experiments, refer to Appendices B and D, respectively.

### 4.1 Experimental Setting

Our experimentation includes three datasets: CIFAR-10 (Krizhevsky & Hinton, 2009), CelebA-HQ (Karras et al., 2018), and ImageNet-1K (Deng et al., 2009), with the latter two resized to $256 \times 256$. For evaluation, we randomly select a subset of 5,000 test images from each dataset and quantitatively measure the reconstruction success with mean squared error (MSE), VGG-based learned perceptual image patch similarity (LPIPS) (Simonyan & Zisserman, 2015; Zhang et al., 2018), and structural similarity index measure (SSIM) (Wang et al., 2004). We note that the employed DM's are not trained on test images.

We report results with respect to $\mu = C/\sigma$, where $C$ denotes the clipping parameter and $\sigma$ the noise multiplier of DP-SGD. It can be interpreted as a signal-to-noise ratio (SNR), where $C$ bounds the signal amplitude and $\sigma$ represents the noise. Analogously to the privacy parameter $\varepsilon$, a *lower* $\mu$ (SNR) makes reconstruction more difficult and, thus, corresponds to a *higher* privacy guarantee. We note that given a specific DP-SGD configuration (number of steps and sampling rate), $\mu$ can be converted to the $(\varepsilon, \delta)$ notion (see Appendix C).

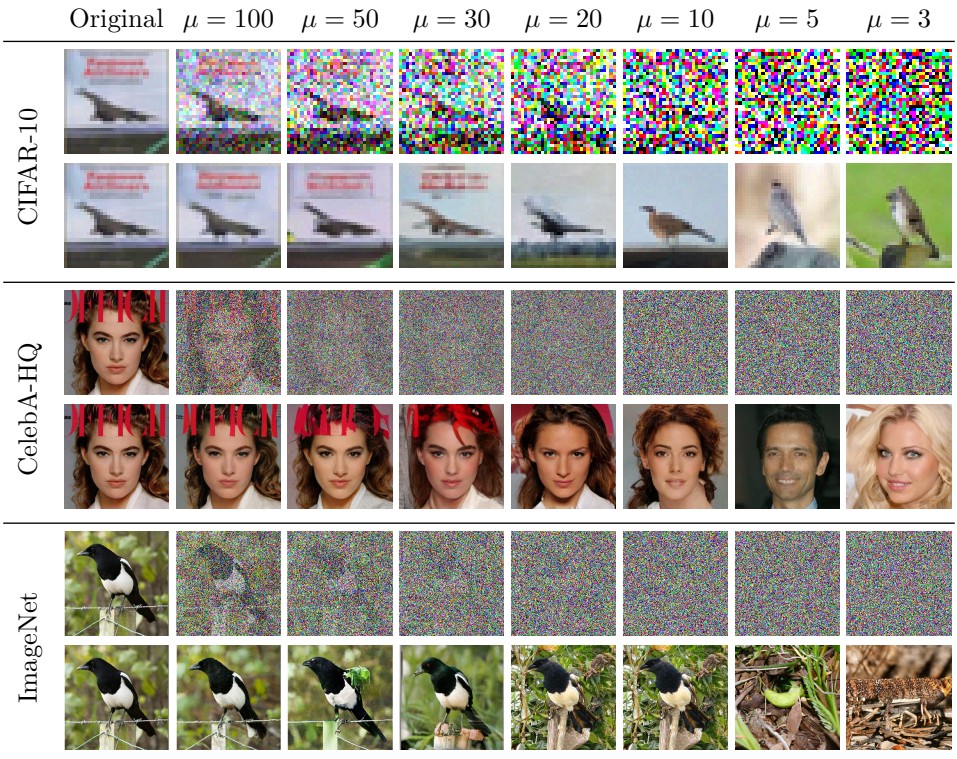

Figure 3: Reconstruction results under DP with respect to $\mu = C/\sigma$. For each dataset, the reconstructed image from the base attack without prior knowledge (*top*) and our DM attack (*bottom*) are shown. The top images also represent the input of our attack.

## 4.2 Reconstruction Success under Different Data Priors

We evaluate privacy leakage under reconstruction attacks with varying levels of prior knowledge about the target data. Our attack incorporates realistic priors learned by diffusion models that capture both general statistics of image features and domain-specific patterns. We compare our approach against the $(0, \gamma)$-ReRo bound, which assumes access to the target image within a prior set of 256 images (Hayes et al., 2023) and the bound of (Ziller et al., 2024b), which assumes no prior knowledge.

The results in Fig. 2 show that our reconstruction error falls between the theoretical bounds: higher than the ReRo bound but lower than Ziller et al.'s uninformed adversary bound[2]. As expected, the ReRo bound is overly pessimistic, assuming a powerful attacker achieving mostly perfect reconstructions—an unrealistic scenario, especially for the challenging ImageNet dataset. Conversely, Ziller et al. are too optimistic and underestimate the threat of a realistic attacker.

A crucial finding emerges when examining reconstruction performance across image scales: As image size increases, the gap between our and Ziller et al.'s results widens, revealing a significant weakness in their method. Images with the same SNR show similar reconstruction difficulty, suggesting that image size should not substantially impact reconstruction success under constant $\mu$. This aligns with both DP and ReRo, which depend on the SNR ratio $C/\sigma$. The discrepancy with Ziller et al.'s findings—where $C\sigma$ is derived for specific error metrics—indicates that directly bounding metrics with limited perceptual relevance may inadequately capture both reconstruction difficulty and actual privacy risk. This also highlights the challenge of formulating an appropriate error function for ReRo that isn't based on matching, as in $(0, \gamma)$-ReRo.

---

[2]In Fig. 2, LPIPS and SSIM for Ziller et al.'s attack converge while the MSE does not. This discrepancy arises not from limitations in their method but from LPIPS and SSIM, which necessitate clipping color values between 0 and 1.

| Original | $\mu = 200$ | $\mu = 150$ | $\mu = 100$ | $\mu = 50$ | $\mu = 30$ | $\mu = 20$ | $\mu = 10$ | $\mu = 5$ |

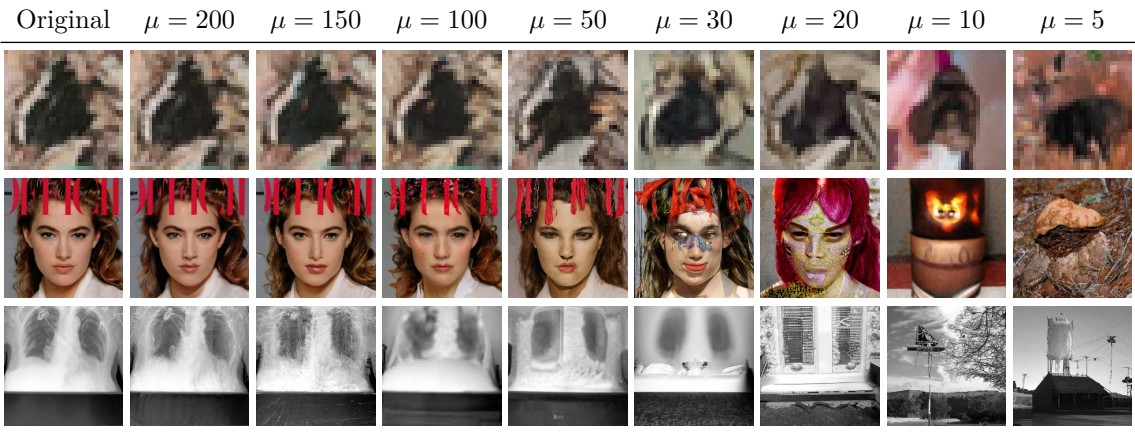

Figure 4: Reconstruction results under distribution shift. The performance of the DM trained on CIFAR-10 and tested on CIFAR-100 (*top*), and the performance of the ImageNet DM on CelebA-HQ (*middle*) and CheXpert (*bottom*) are shown.

Regarding our reconstruction success, a "phase transition" becomes apparent. For $\mu \leq 5$, the similarity of our reconstructions to the original converges to the average similarity of test images (dashed line in Fig. 2), indicating the diffusion model generates plausible but unrelated images from the learned distribution.

The qualitative results in Fig. 3 and Appendix E demonstrate the strong performance of our DM-based attack against the DP-SGD defense. While the base attack yields noisy images, the DM successfully recovers substantial original image content. Notably, we observe an additional phase transition at $\mu = 20$, where reconstructed images start deviating from the original while still sharing similar high-level attributes such as dataset class, image color, or gender. Additionally, as observed in Fig. 2, for $\mu \leq 5$, the reconstructions become unrelated to the original, indicating good privacy protection.

### 4.3 Reconstruction Success under Distribution Shift

Previously, we assumed the adversary has access to training data with the same underlying data distribution as the target (test) data, which enables them to learn a very strong data prior. To investigate how prior knowledge quality affects reconstruction success, we examine scenarios where the data prior does not stem from the same distribution, *i.e.*, an out-of-distribution prior, which is weaker than in-distribution priors. We perform this experiment in three settings: (1) The DM is trained on CIFAR-10 and is used to reconstruct test images from CIFAR-100 (Krizhevsky & Hinton, 2009). These datasets are very similar, differing primarily in class number and diversity. (2) An ImageNet DM reconstructs CelebA-HQ face images, and (3) the ImageNet DM reconstructs grayscale chest X-ray images from the CheXpert dataset (Irvin et al., 2019). Intuitively, the greater the discrepancy between training and test data, the larger the distribution shift.

The results in Figs. 4 and 5 show a clear trend: larger distribution shifts (weaker data priors) lead to decreased reconstruction success. This is particularly evident from the shift in the privacy guarantee ($\mu$-value) at which the similarity of the reconstructions surpasses the average similarity of the test datasets (dashed line in Fig. 5). Irrespective of the error function, this serves as a good indicator for less-than-useful reconstructions, which are more similar to the training data than the target images.

Our findings demonstrate the significant impact of data distribution shift on the reconstruction performance of DMs, especially in high privacy regimes. However, our method yields reasonable reconstructions for low privacy guarantees even in scenarios with significant distribution shifts.

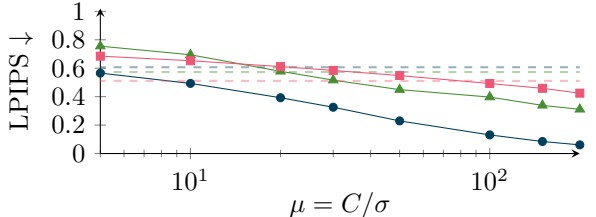 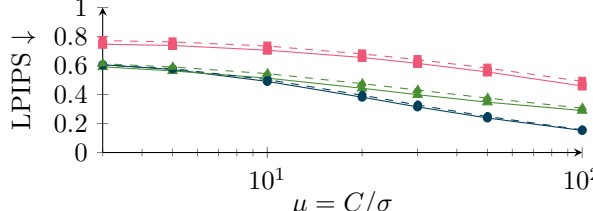

Figure 5: Reconstruction success under distribution shift. The performance of the DM trained on CIFAR-10 and tested on CIFAR-100 (*blue* ●), and the ImageNet DM tested on CelebA-HQ (*green* ▲) and CheXpert (*red* ■) are shown. The *dashed* line represents average similarity between test images of the datasets (same color). The results show the significant influence of distribution shift between the data prior and the reconstruction target.

Figure 6: Reconstruction success estimated from average similarity between multiple DDPM samples of CIFAR-10 (*blue* ●), CelebA (*green* ▲), and ImageNet (*red* ■) compared to the true success obtained by computing the average similarity between reconstructions and the original images (*dashed* line). The closeness between same colored lines shows that the original image is not required to estimate the reconstruction success well.

## 4.4 Estimating Reconstruction Success without Target Access

DMs always generate a candidate reconstruction, even when the perturbed image lacks information for reconstruction. This implies that the resulting reconstructions may differ from the target images. While this is useful for data owners and practitioners who can directly compare the features of the reconstructions with the original images, adversaries lacking access to the original image (reconstruction target) may struggle to infer which features are made up by the DM.

We propose that adversaries can overcome this challenge by generating multiple candidate reconstructions using the probabilistic generation process of, *e.g.*, DDPMs and assess which features remain consistent across reconstructions. Such features are most likely to originate from the reconstruction target. This is analogous to a *maximum a posteriori* attack, where the mode of the empirically generated images is computed.

Fig. 6 shows the average pairwise similarity between five DDPM generations from each of the 5,000 noisy images under different privacy levels, providing insights into estimating the reconstruction success using such an approach. It shows that, for all datasets, the true reconstruction success (dashed lines in Fig. 6) can be estimated well without access to the original image. Qualitative results in Supplementary Fig. 15 illustrate the shared features among different generations. In our example, gender can be inferred until $\mu = 5$, and the hair color remains consistent until $\mu = 20$.

These findings highlight that visual insights from DMs' reconstructions hold value not only for data owners comparing reconstructions with the original images but also for adversaries who only have access to the noisy image and multiple generations.

## 4.5 Reconstruction Success under Weaker Attack Assumptions

While our previous experiments assumed ideal attack conditions to align with theoretical bounds, we now evaluate our method's effectiveness in a practical training scenario where adversaries can only access accumulated mini-batch gradients rather than per-sample gradients. This setting aligns with real-world DP-SGD implementations and follows the attack methodology of Fowl et al. (2022). The key challenge lies in extracting individual sample information from aggregated gradients.

Our experimental setup employs mini-batches of size 64 and a ResNet-9 architecture (Klause et al., 2022) augmented with an imprint layer (Fowl et al., 2022). The imprint layer's parameters are carefully tuned to separate individual activations within the accumulated gradient using a binning technique (128 bins). For reconstruction, we process the binned weight and bias gradients according to Eq. (6), instead of relying on

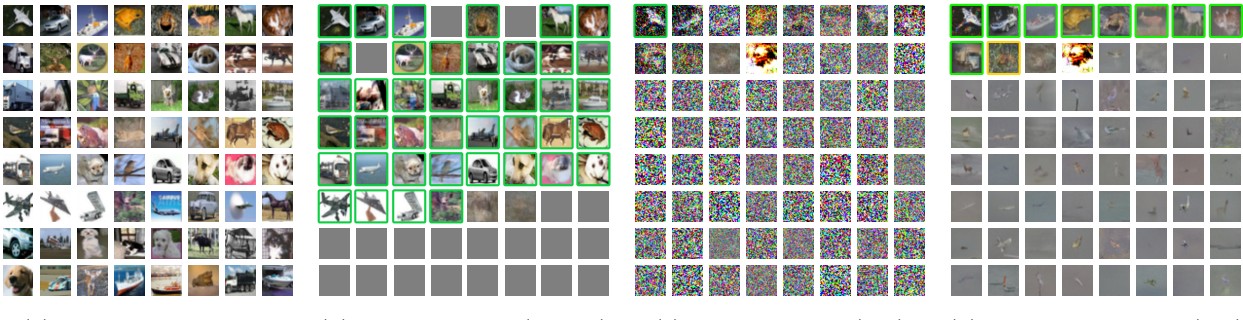

(a) Ground truth batch    (b) Reconstructed (no DP)    (c) Reconstructed (DP)    (d) DM reconstructed (DP)

Figure 7: Reconstruction results on ResNet-9 training using the attack of Fowl et al. (2022) under: (b) no privatization, (c) DP-SGD, and (d) DP-SGD including our DM attack strategy. DP-SGD is applied with $\mu = 1000$. Cells are sorted by reconstructions success of (d). Empty cells (gray) indicate failed reconstructions, while green borders highlight successfully reconstructed images matching the original training data.

the clipping factor $\lambda$ approach used in prior experiments. This produces reconstructions with noise following a Gaussian ratio distribution, though the precise distribution may deviate due to the binning process.

Our attack pipeline consists of four key steps: (1) apply binning to separate individual contributions from the observed accumulated gradient, (2) divide weight and bias gradients for each bin to yield the (perturbed) reconstruction, (3) determine the reverse diffusion starting point $t_{\text{start}}$ by estimating the noise variance using `scikit-image`, and (4) apply our DM reconstruction method to each bin. This approach enables simultaneous recovery of all images in the batch from a single accumulated gradient.

Results in Fig. 7 demonstrate the effectiveness of our attack under these non-ideal conditions. Without DP, most images are reconstructed with high fidelity (Fig. 7b). While DP-SGD protection initially reduces most bins to noise patterns (Fig. 7c), our DM-based method successfully recovers many recognizable features from these protected versions (Fig. 7d). This strongly supports the effectiveness of data priors in reducing DP's effects. Finally, it is important to note that $\mu$ can be significantly larger in this scenario due to other non-zero gradients in the ResNet architecture that are not used for reconstruction but add to the gradient norm. This effect decreases the signal for the relevant gradients while the noise scale remains constant, further challenging the reconstruction process.

## 5    Discussion

Our investigation into real-world data priors reveals a significant gap between theoretical reconstruction bounds and empirical attack outcomes. We demonstrate that the strength of the data prior substantially influences the reconstruction success, positioning our attack between existing bounds. This finding highlights both the importance and challenge of incorporating realistic data priors into formal privacy guarantees.

While our work primarily provides empirical evidence of enhanced reconstruction capabilities through diffusion models (DMs), establishing theoretical bounds for reconstruction attacks under such data priors remains an open challenge. The key difficulty lies in capturing the semantics of learned representations and their relationship to the private training data in a formal framework. While threat models assuming no prior (Ziller et al., 2024b) remain valuable where priors are unavailable, incorporating prior knowledge could substantially improve bound accuracy. We also recognize the flexibility of the ReRo bound in formalizing different data priors. Nevertheless, addressing challenges related to defining an appropriate prior $\pi$ and error functions $\rho$ will be crucial for its effective implementation. The development of theoretical foundations for realistic data priors, such as those learned by DMs, represents an important direction for future research.

Furthermore, we find that DMs excel at extracting information from heavily perturbed images beyond human visual capabilities. Our method substantially improves the reconstruction outcomes of previous methods (Fowl et al., 2022; Boenisch et al., 2023b; Ziller et al., 2024b) with a simple post-processing step. Given the

widespread availability of pre-trained DMs across various data distributions, it is reasonable to assume that adversaries can profit from their capabilities. This accessibility broadens the scope of potential adversaries who could utilize such techniques, emphasizing the urgency for robust defenses to counter such threats.

However, this same capability presents an opportunity: DMs can serve as powerful tools for visualizing reconstruction risk in privacy audits. Our reconstructions effectively capture the residual information after privatization, offering intuitive insights into privacy leakage that complement formal DP guarantees. Unlike abstract privacy parameters that are challenging to interpret for non-experts (Cummings et al., 2024), our approach offers tangible means of visualizing privacy leakage, thereby facilitating communication with stakeholders and enhancing their comprehension of privacy in machine learning. For instance, when reconstructions preserve class information while altering low-level features, it suggests that low-level features are privatized, whereas the class information can be disclosed (Sec. 4.4). This practical approach to privacy auditing aligns with recent developments in heuristic auditing methods (Steinke et al., 2024). We emphasize, however, that our method is intended as a communication tool and does not provide theoretical guarantees.

Finally, since DP ensures consistent mathematical privacy guarantees regardless of prior knowledge—even under ideal priors—our findings suggest an interesting practical consideration: the standard noise levels in mechanisms like DP-SGD may be excessive. We demonstrate that prior-based post-processing of privatized gradients can increase the utility of data reconstruction attacks. The same approach can be used to get improved model utility in DP-SGD training by post-processing gradients before the model update, while obtaining the same mathematical guarantee. This insight aligns with recent advances in gradient denoising techniques (Nasr et al., 2020; Zhang et al., 2024), suggesting promising directions for future research.

## 6 Conclusion

Our work empirically shows the critical role of data priors in reconstruction attacks, revealing limitations in current theoretical bounds. This gap between theory and practice highlights the need for reconstruction bounds that better capture real-world adversarial capabilities. We show both the threat and utility of DMs in privacy contexts: while they enhance reconstruction attacks, they also enable intuitive privacy auditing that bridges the gap between theoretical guarantees and practical understanding. Future work should focus on developing more adaptive privacy metrics and defenses that can address realistic capabilities of adversaries.

**Broader Impact Statement**

This work proposes a data reconstruction attack capable of disclosing sensitive information from real-world ML models. While our method can be used maliciously, we use it to highlight how to defend against data reconstruction attacks using privacy methods like DP and how our attack can be utilized as a tool for non-experts to select sufficient privacy guarantees. Furthermore, we only utilize publicly available images, thereby not exposing data that is not already available.

**Acknowledgments**

KS and DR received support from the Bavarian Collaborative Research Project PRIPREKI of the Free State of Bavaria Funding Programme "Artificial Intelligence – Data Science".

AZ and DR received support from the German Ministry of Education and Research and the Medical Informatics Initiative as part of the PrivateAIM Project (grant no. 01ZZ2316C).

JK received support from the European Union under Grant Agreement 101100633 (EUCAIM). Views and opinions expressed are however those of the author(s) only and do not necessarily reflect those of the European Union or the European Commission. Neither the European Union nor the granting authority can be held responsible for them.

MK was supported by the DAAD programme Konrad Zuse Schools of Excellence in Artificial Intelligence, sponsored by the Federal Ministry of Education and Research.

SL received support from the Research and Development Program Information and Communication Technology Bavaria, DIK0444/03.

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

# A    Approximating the Clipping Factor $\lambda$

Recall our assumption that the adversary has knowledge about the exact value of the clipping factor $\lambda = \max(||\boldsymbol{x}||_2/C, 1)$ in Eq. (7). In this section, we demonstrate the simplicity of yielding a good approximation of $\lambda$ using trial-and-error.

As an example, we take an image from the CIFAR-10 dataset and assume $C = 1$ (a standard value in DP-SGD practice (Ponomareva et al., 2023)) and $\mu = C/\sigma = 30$. The example image has a L2-norm of $||\boldsymbol{x}||_2 = 27.24$, thus, it will be clipped and $\lambda = ||\boldsymbol{x}||_2/C = 27.24$.

The first step of an adversary could be to set a value range for $\lambda$. Given the standard range of color values $x_i \in [0, 1]$, the maximum L2-norm of a flattened image $\boldsymbol{x} \in \mathbb{R}^{HWD}$ is $\sqrt{HWD}$, in this case, $(||\boldsymbol{x}||_2)_{\max} = \sqrt{32 \cdot 32 \cdot 3} = 55.43$ and, therefore, $\lambda \in [1, 55.43]$. Now, without further assumptions, the adversary can repeat the reconstruction process with different values for $\lambda$ and select the best result. Figure 8 shows some example results of the trial-and-error approach.

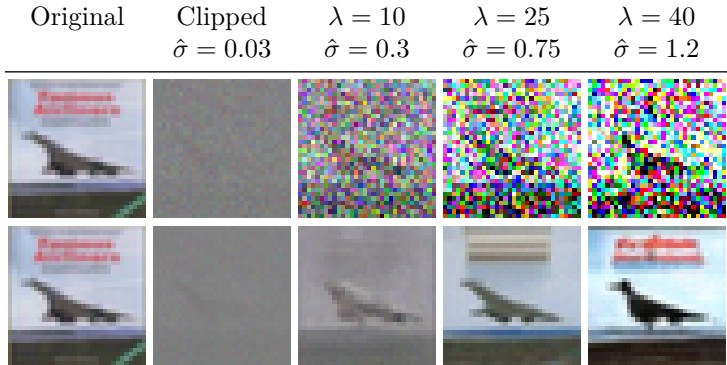

Figure 8: Reconstruction results for different approximations of $\lambda$. The (scaled) perturbed image (*top*) and the DM's reconstruction (*bottom*) are shown. Additionally, the change in the standard deviation of the noise $\hat{\sigma} = C\sigma\lambda$ resulting from rescaling is given.

# B    Experimental Details

**Models.**    For the CIFAR-10 and CelebA-HQ datasets, we utilize the diffusion models and exponential moving average (EMA) checkpoints from Ho et al. (2020). These checkpoints achieve a validation Fréchet Inception Distance (FID) (Heusel et al., 2017) of 3.17 for CIFAR-10, the FID for CelebA-HQ was not reported. For the ImageNet dataset, we employ the unconditional DM of Dhariwal & Nichol (2021), which achieves a validation FID of 12.00. All DMs are based on U-Net architectures (Ronneberger et al., 2015) and PixelCNN++ (Salimans et al., 2017).

**Frameworks.**    We use the `Diffusers` library (von Platen et al., 2022) (based on `PyTorch` (Paszke et al., 2019)) to leverage state-of-the-art pre-trained DMs for implementing our reconstruction attack.

**Compute Reconstruction Performance.**    To evaluate the reconstruction performance of the considered attacks, we asses the average similarity between the reconstructed test images and their original counterparts. First, we execute the reconstruction attack proposed by Ziller et al. (2024b) on DP-SGD (as described in Sec. 2), obtaining their reconstruction performance. Then, we post-process the noisy images generated by Ziller et al.'s attack using our proposed DM method (as described in Sec. 3), representing our attack's reconstruction performance.

For the ReRo lower bound, we implement the prior-aware attack proposed by Hayes et al. (2023) under identical DP-SGD settings and architectures as Ziller et al. (2024b). We compute reconstructions by matching the noisy and clipped gradients from DP-SGD with the clipped gradients derived from a prior set comprising

256 candidate images using the dot product. The resulting matched images are then considered as reconstructions and used to compute the similarity. Intuitively, successful matching by the ReRo attack results in perfect reconstructions.

## C  Interpreting $\mu$ for Standard DP-SGD Configurations

As discussed in Sec. 4.1, we report results using the signal-to-noise ratio (SNR), defined as $\mu = C/\sigma$, where $C$ is the clipping parameter and $\sigma$ is the noise multiplier in DP-SGD. This metric provides a privacy measure that is independent of specific training hyperparameters, enabling generalization of our results across various training configurations.

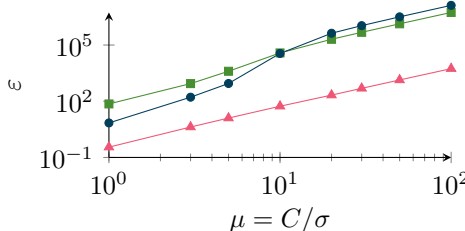

In DP, privacy guarantees are typically expressed in the $(\varepsilon, \delta)$-DP framework. For a fully defined DP-SGD configuration (including the number of steps $T$, sampling probability $p$, and $\delta$), $\mu$ can be converted into an $(\varepsilon, \delta)$ pair by using advanced privacy accountants (Mironov, 2017; Gopi et al., 2021).

To facilitate comparisons with the $(\varepsilon, \delta)$ notion, we present $\varepsilon$ estimates for representative DP-SGD configurations in Fig. 9. These values are illustrative and were not directly used in our experi-

Figure 9: Conversion of $\mu$ to $(\varepsilon, \delta)$-DP for typical DP-SGD configurations for CIFAR-10 (*blue* ●) and ImageNet (*green* ■). Additionally, we adopt a worst-case configuration (*red* ▲).

ments, as our setup does not rely on specific configurations. For CIFAR-10 with $N = 50,000$ training samples, we adopt the parameters from (Klause et al., 2022), using $T = 2,400$ steps, a sampling probability of $p = 1,024/50,000$, and $\delta = 10^{-5}$. For ImageNet ($N = 1,281,167$), we employ the fine-tuning setting from (Berrada et al., 2023), with $T = 1,000$, $p = 262,144/1,281,167$, and $\delta = 8 \cdot 10^{-7}$. Additionally, we consider a worst-case scenario where an adversary modifies the hyperparameters to $T = 1$ step and a batch size of 1, resulting in $p = 1/50,000$.

Our results reveal large differences in privacy guarantees across these configurations, underscoring the substantial impact of training hyperparameters on privacy protection. These findings support our choice of using $\mu$ as an independent, robust metric in our analysis.

## D  Ablation Experiments

In this section, we conduct a series of ablation experiments to assess the performance of our reconstruction attack under different settings and assumptions. Each ablation experiment evaluates the average similarity between the reconstructed images and the original images using CIFAR-10 and LPIPS. In all figures, the dashed line represents the average similarity of test images.

**Privacy Leakage from Re-Identification.** In this experiment, we evaluate the capabilities of an adversary using our method for re-identification. For this, we employ the matching strategy introduced by Hayes et al. (2023) (see Sec. 2.1) and match the reconstructed image with the most similar image from a prior set using LPIPS and compute the ratio of correct matches. We compare our matching success with the $(0,\gamma)$-ReRo bound.

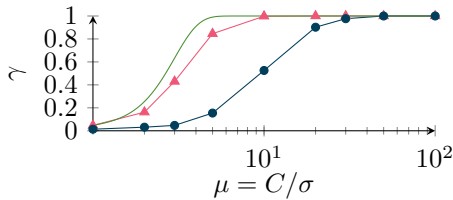

Figure 10: Image matching success $\gamma$, with prior set size of 256 using our reconstructed images (*blue* ●) compared to the ReRo lower (*red* ▲) and upper bound (*green*).

The results in Fig. 10 corroborate our expectation that our method achieves lower matching success than the ReRo bound. This difference can be attributed to our attack solely assuming a general data prior, while ReRo assumes access to the full underlying dataset. Consequently, our method relies on the DM-based reconstruction of the image, which may drop some information in the generation process. However, despite this limitation, our matching perfor-

mance is notably close to the ReRo bound, indicating our attack's ability to recover unique features even under strong perturbations. Once again, we observe that $\mu \leq 5$ serves as a threshold beyond which our attack cannot recover any unique features.

**Comparison between DDIM and DDPM Generation.** Recall that we employ the deterministic generation process of DDIMs to enforce data consistency. In this experiment, we evaluate the effect of this design choice on reconstruction performance by comparing DDIM generation with the probabilistic generation process of DDPMs, which is usually used in implementations of DMs.

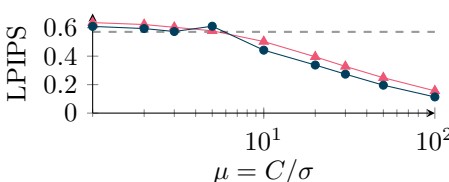

Figure 11: DDIM (*blue* •) and DDPM (*red* ▲) generation performance.

The results in Fig. 11 show improved performance across various privacy levels with DDIM sampling, suggesting that DDIMs exhibit greater consistency and remain closer to the original image throughout the generation process.

**Unknown Noise Variance.** In our main experiments, we assume that the adversary knows the variance of the noise in the privatized image. However, this assumption may not always hold true in practical scenarios. In this experiment, we assess the impact of unknown noise variance on our reconstruction performance. For this, we approximate the noise variance using the wavelet-based implementation in `scikit-image` (van der Walt et al., 2014) (`restoration.estimate_sigma`), which is described in Section 4.2 of (Donoho & Johnstone, 1994).

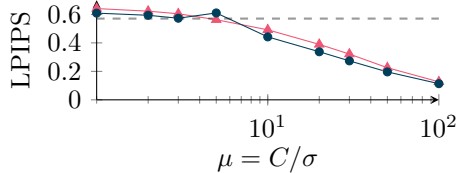

Figure 12: Performance under known noise variance (*blue* •) and noise variance estimation (*red* ▲).

The results in Fig. 12 show only a slight decrease in reconstruction performance, indicating that the noise variance can be accurately estimated and that our attack is robust against estimation errors.

**Denoising without Learned Data Priors.** In this experiment, we assess the effectiveness of structural image priors that only capture patterns inherent in images, and do not approximate a specific data distribution. For this, we employ traditional denoising methods based on wavelet transformation (Chang et al., 2000) and BM3D (Dabov et al., 2007) and compare their performance with our DM method approximating the underlying data distribution.

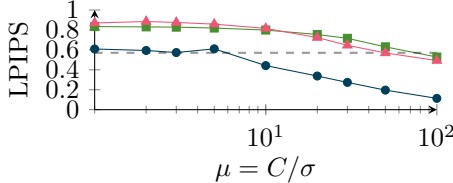

Figure 13: Performance of traditional denoising methods based on wavelet transformation (*green* ■) and BM3D (*red* ▲) compared to our DM method (*blue* •).

The results in Fig. 13 show the limitations of traditional denoising methods when confronted with large noise perturbations. Specifically, the results reveal a large performance difference across all privacy levels.

## E    Additional Reconstruction Results

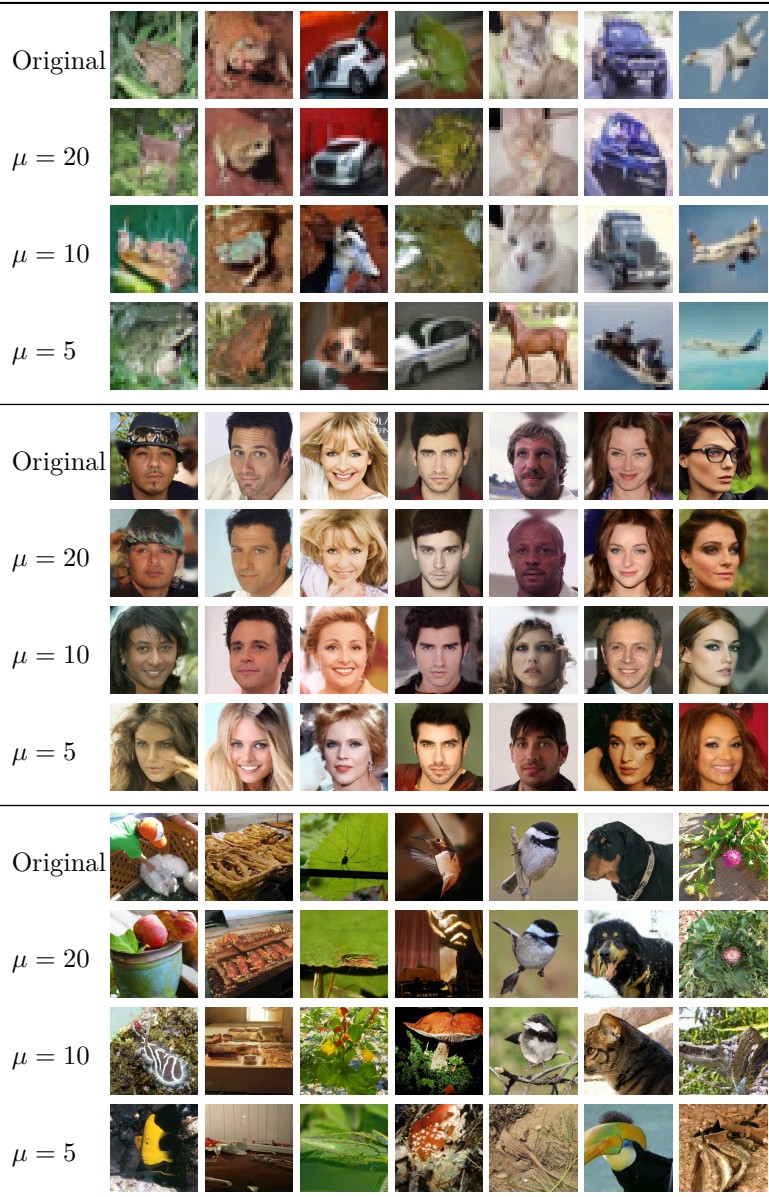

Figure 14: Reconstruction results of our DM attack with respect to $\mu = C/\sigma$ for CIFAR-10 (*top*), CelebA-HQ (*middle*), and ImageNet (*bottom*).

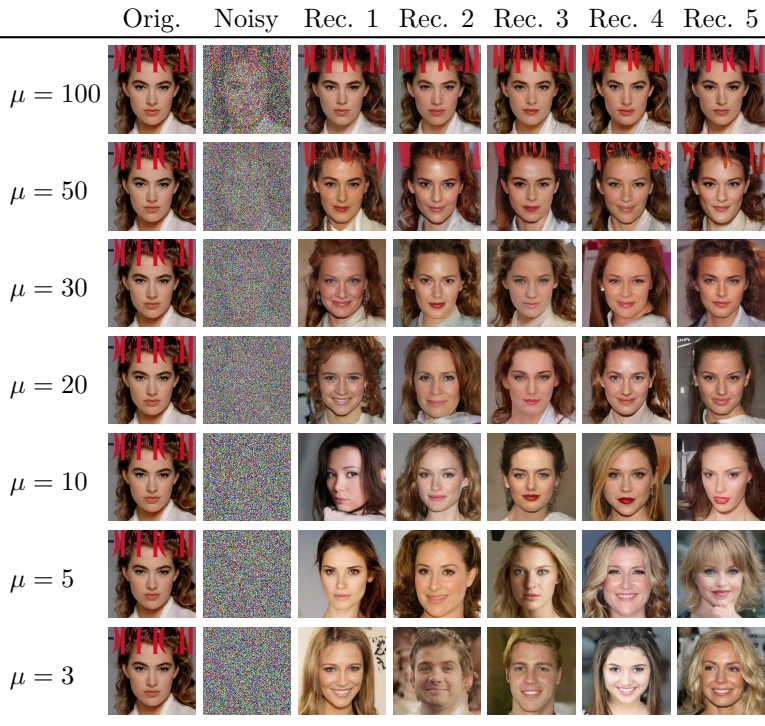

Figure 15: DDPM reconstruction results with respect to $\mu = C/\sigma$ for a CelebA-HQ image. For each $\mu$-value, the original image, the noisy image, and five reconstructions from the noisy image are shown. We observe that lower $\mu$ (SNR) lead to larger deviations between generations. The adversary is interested in the features that stay consistent across generations, as these likely originate from the original image. For example, hair color stays consistent until $\mu = 20$, and gender can be inferred until $\mu = 5$. This shows that the visual insights from reconstructions of DMs are not only valuable for data owners who can compare the reconstruction with the original image, but also for adversaries who only have access to the noisy image and the ability to compare different generations.

