# OpenReview forum: "Visual Privacy Auditing with Diffusion Models"
_TMLR — Accepted by TMLR_

### Review · Reviewer_2F3d · 2024-11-25

**Summary Of Contributions:**

In this paper, the authors investigate how well prior theoretical bounds on image reconstruction attack success under differential privacy (DP) hold under more realistic data priors. Namely, the authors reference two existing papers which provide theoretical reconstruction attack success with opposing levels of adversarial assumptions. Then, this paper introduces a new attack for reconstructing images from privately trained models using diffusion models. The authors then use this novel attack to show that their realistic data prior assumptions + attack settle between the two bounds provided by prior work. This shows that assumptions on prior data significantly impact reconstruction success, which limits the practicality of existing theoretical bounds. Finally, the authors state that their diffusion-based private reconstruction attack can thus be used as an effective auditing tool for privacy leakage (with a visualization component).

**Audience:**

Yes

**Claims And Evidence:**

Yes

**Requested Changes:**

- can the authors include classifier training details used for their three datasets? It is hinted that resnet-9 was used, but it would be good to know the full training details + testing accuracy when applying DP
- it is mentioned that increasing image size furthers the gap between Ziller et. al’s bound and the DM attack, but this finding isn’t *quantifiably* clear from the manuscript. Is there a way to include a measurement of the increasing gap between Ziller et al’s bound and the DM attack?

**Strengths And Weaknesses:**

[strengths]
- accuracy of reconstruction success can be approximated even without ground-truth images
- necessary and well-measured comparison between theoretical bounds with assumptions and attacks that use practical priors
- novel and interesting usage of diffusion models to reconstruct images, very clear empirical results on its success compared to prior work

[weaknesses]
- diffusion models blur the line between “successful” and “unsuccessful” reconstruction attack, since the reconstructed image often looks like a real image, though this is partially addressed by success approximation

---

> ### Author Response · Authors · 2025-01-15
> **Response to Reviewer 2F3d**
>
> Thank you for your positive and constructive review! Below, we address the weaknesses and requested changes you highlighted.
>
> # Weaknesses
>
> > Diffusion models blur the line between “successful” and “unsuccessful”
>
> We agree with this observation. However, as noted, we address this challenge through our empirical validation showing that attack success can be reliably approximated by comparing multiple generations/reconstructions. Our quantitative metrics demonstrate strong correlation with the ground truth of reconstruction quality.
>
> # Requested Changes
>
> > Including full training details and testing accuracy
>
> Thank you for this suggestion. We adopted the training setting from [1]. This choice was intentional to ensure that our results are consistently comparable with those.
>
> Regarding testing accuracy and more detailed training parameters, we note that these either have no direct impact on attack success (e.g. accuracy and learning rate) or substantially affect the privacy guarantee (e.g. number of steps $T$, sampling probability $p$, or $\delta$). Such dependencies reduce the generality of results. To address this, we used the signal-to-noise ratio (SNR), defined as $\mu=C/\sigma$, instead of $\varepsilon$, when reporting results. Unlike $\varepsilon$, which depends on specific hyperparameters, $\mu$ provides a more generalizable privacy metric.
>
> We acknowledge however, that specific hyperparameter settings are interesting. To address this, we included concrete examples in Appendix C of the revised manuscript. These examples highlight how hyperparameters like $T$ and $p$ substantially affect privacy guarantees.
>
> > Quantifying the gap between Ziller et al.’s bound and the DM attack
>
> To quantify the gap between Ziller et al.'s bound and our DM attack at different image scales, we can examine, e.g., the MSE values in Fig. 2 at $\mu=100$. For $32\times 32$ CIFAR-10 images, the MSE gap between methods is approximately 0.018. This gap substantially widens to approximately 1.696 for $256\times 256$ CelebA-HQ images - a 94x increase. This directly demonstrates how larger image sizes increase the difference between the methods.
>
> The theoretical basis for this increasing gap stems from Ziller et al.'s bounds yielding $C\sigma$ instead of the standard $C/\sigma$ ratio (as discussed in Sec. 4.2). While a more rigorous mathematical quantification isn't feasible due to our attack's reliance on black-box diffusion models, the empirical results clearly demonstrate the gap across different $\mu$ values. To help readers to better quantify these differences, we have improved Fig. 2's readability by increasing the number of tick marks in the MSE plots.
>
> # References
>
> [1] Alexander Ziller, Anneliese Riess, Kristian Schwethelm, Tamara T. Mueller, Daniel Rueckert, and Georgios Kaissis. Bounding reconstruction attack success of adversaries without data priors, 2024b.

---

> > ### Comment · Reviewer_2F3d · 2025-01-22
> > **Thank you for rebuttal**
> >
> > I would like to thank the authors for their responses. My comments have been addressed.

---

### Review · Reviewer_rCx3 · 2024-12-08

**Summary Of Contributions:**

The paper develops a novel data reconstruction attack using diffusion models (DMs). Specifically, the contributions are:

1. The observation that real-world data priors significantly affect the success of reconstruction attacks, challenging current theoretical bounds.

2. The introduction of a new attack incorporating DMs to enhance reconstruction of images protected by Differential Privacy (DP) mechanisms, such as DP-SGD.

3. The observation/proposal to use DMs as heuristic tools for visualizing privacy leakage.

**Audience:**

Yes

**Claims And Evidence:**

No

**Requested Changes:**

1. Could you please convert the μ to ε for Figure 2?

2. There are works that either use DP-GANs [1,2,3] or synthesize images with DP guarantees that are visually different from the private images[4]. Please explain how this method is different from them and compare with them in terms of performance and privacy. Specifically, currently the paper evaluates the reconstruction performance of the private images for a model trained with DP-SGD. How the DM generated images are better than the GAN-generated images? What are the advantages of using the DM? Furthermore, [4] shows that a model can be trained using visually different synthetic and achieve comparable utility as the private images. Please explain more the concept that of visual privacy that the paper proposes and how this is different from other SOTA works.

[1] Liyang Xie, Kaixiang Lin, Shu Wang, Fei Wang, and Jiayu Zhou. Differentially private generative adversarial
network. arXiv preprint arXiv:1802.06739, 2018

[2] Frederik Harder, Kamil Adamczewski, and Mijung Park. Dp-merf: Differentially private mean embeddings
with randomfeatures for practical privacy-preserving data generation. In International conference on
artificial intelligence and statistics, pp. 1819–1827. PMLR, 2021

[3] Shun Takagi, Tsubasa Takahashi, Yang Cao, and Masatoshi Yoshikawa. P3gm: Private high-dimensional
data release via privacy preserving phased generative model. In 2021 IEEE 37th International Conference
on Data Engineering (ICDE), pp. 169–180. IEEE, 2021

[4]Soufleri, Efstathia, Deepak Ravikumar, and Kaushik Roy. "DP-ImgSyn: Dataset Alignment for Obfuscated, Differentially Private Image Synthesis." Transactions on Machine Learning Research.

Minor comment:

 Is any way to establish theoretically your proposal?Note that the paper demonstrates the experimental results and additional theory would add extra value

**Strengths And Weaknesses:**

Strengths:

1. The paper proposes a new attack using DM for analyzing privacy risks and offers practical insights into the limitations of DP guarantees.

2. Experimental evaluation of the proposed framework on CIFAR-10, CelebA-HQ, and ImageNet datasets.

3. The paper is well-written and structured.

Weaknesses:

Please see the requested changes below.

---

> ### Author Response · Authors · 2025-01-15
> **Response to Reviewer rCx3**
>
> Thank you for your positive feedback! We have addressed the requested changes you outlined below.
>
> > 1. Conversion of $\mu$ to $\varepsilon$ in Fig. 2
>
> We appreciate this suggestion. We chose the signal-to-noise ratio $\mu$ as our primary metric because it enables universal insights that generalize across different training configurations. Unlike $\varepsilon$, which varies substantially based on hyperparameters such as the number of training steps $T$ and sampling probability $p$, $\mu$ provides a configuration-independent measure of privacy strength. However, we understand the practical importance of $\varepsilon$ for practitioners. To address this, we have added concrete examples of $\varepsilon$ values under typical DP-SGD configurations to Appendix C of the revised manuscript. These examples demonstrate how hyperparameters like $T$ and $p$ substantially influence $\varepsilon$, supporting our decision of using $\mu$.
>
> > 2. Works on DP image generation
>
> Thank you for bringing up these interesting works. We note that our work addresses a fundamentally different problem than the cited papers. While these works focus on generating private synthetic data directly from sensitive datasets for training purposes, our research examines the inverse problem: reconstructing sensitive images from observed private gradients during training. The key distinction is in directionality - where these works develop methods to create privacy-preserving data from source data, we analyze how an attacker might reconstruct source data from privatized observations.
>
> > 3. Theoretical results
>
> As highlighted in the paper's discussion section, developing theoretical bounds for reconstruction attacks under realistic data priors remains an open challenge and an important direction for future work. We plan to explore these theoretical aspects in follow-up research.

---

> > ### Comment · Reviewer_rCx3 · 2025-01-28
> >
> > I would like to thank the authors for their responses. My first comment has been adequately addressed. However, regarding comment 2, I still have some doubts. The works I cited earlier focus on generating synthetic data with differential privacy (DP) guarantees that are visually dissimilar to the private data. This ensures that an attacker cannot reconstruct the private data from the synthetic data. I recommend including a paragraph in the related work section discussing these works and explicitly clarifying how this proposal differs from them. Finally, regarding comment 3, I suggest explicitly mentioning it in the revised discussion section of the paper. Thank you.

---

> > > ### Author Response · Authors · 2025-01-29
> > > **Response to Reviewer rCx3**
> > >
> > > We thank the reviewer for the follow-up comments. While we appreciate the suggestion to discuss works on DP synthetic data generation, we respectfully maintain that these works address a fundamentally different problem than our paper. Including them in our related work section could potentially mislead readers about the scope and goals of our paper. The key differences are:
> > >
> > > 1. Goal: The cited works develop defense mechanisms to prevent privacy leakage through synthetic data generation, while our work analyzes the vulnerability of existing privacy mechanisms through reconstruction attacks.
> > > 2. Setting: The cited works focus on privatizing raw data before any downstream use, while our work specifically examines privacy leakage during the model training process.
> > > 3. Threat Model: The cited works consider an adversary with direct access to raw (synthetic) data, while our work analyzes scenarios where an adversary only has access to privatized intermediate computations (privatized gradients) during training.
> > >
> > > Regarding comment 3, we appreciate the suggestion and have revised our discussion section accordingly to explicitly address this point.

---

### Review · Reviewer_j7Cv · 2025-01-08

**Summary Of Contributions:**

The paper investigates the usage of diffusion models for gradient inversion attacks on vision models during differentially private training. In this setting, the diffusion model has prior knowledge of the data distribution and, therefore, has more capability to attack the DP training successfully. The authors argue that the level of data privacy protection by DP is not accurate, considering the possible prior knowledge of the data distribution gained by the diffusion model. The paper designs a series of numerical experiments to demonstrate the ability of diffusion models to recover the original image from the privatized gradient under different SNRs.

**Audience:**

Yes

**Claims And Evidence:**

No

**Requested Changes:**

Please check the weaknesses above.

**Strengths And Weaknesses:**

Strength:
1. The paper proposes using diffusion models for data reconstruction in DP training.
2. The paper conducts a series of numerical experiments on using diffusion models for image recovery from privatized gradients for different SNRs and compares them with the algorithms without using diffusion models.

Weakness:
1. It is unclear how the privatized image $x_{priv}$ is obtained. In DP-SGD, the attacker can only get the **averaged** gradient instead of the individual gradient of one sample. Therefore, the adv. problem eq(7) is not a valid setting at first glance. Please provide a more in-depth discussion on how to obtain the input $x_{priv}$.
2. The paper's section 4.5 assumes only using mini-batch SGD. This setting is more aligned with the practical DP training. The authors should provide a detailed discussion of the algorithm and experiment setting for this case.
3. In section 4.1, the author mentioned that $\mu$ can be translated to $(\epsilon,\delta)$. More rigorous discussion should be given. Specifically, the authors should provide a few examples of $\mu$ for typical DP training settings on the tested datasets.
4. The paper should discuss more related work using diffusion models for privacy attacks, e.g., [[R1]](https://arxiv.org/pdf/2411.03053), [[R2]](https://arxiv.org/pdf/2407.05285)
5. The input reconstruction requires the input layer to be fully connected layer. For general convolutional layers, is there a way to recover the privatized input from the per-sample gradient?

---

> ### Author Response · Authors · 2025-01-15
> **Response to Reviewer j7Cv**
>
> Thank you for your detailed comments! Below, we address the concerns and weaknesses you have highlighted.
>
> > 1. Obtaining the privatized image for Eq. (7)
>
> Please note that Eq. (7) represents a valid setting under our threat model. For simplicity, it assumes that the adversary has already performed the necessary steps to extract the image from the observed gradient. Consistent with prior work, our threat model allows the adversary to set hyperparameters, such as using a batch size of 1, to directly access per-sample gradients and thus the perturbed image. Additionally, in Section 4.5, we explore a weaker attack scenario where the adversary extracts the image from accumulated gradients using a binning technique. We have revised the manuscript to clarify this simplification in representation and the steps involved in gradient extraction (see Sec. 3.1).
>
> > 2. Details on mini-batch SGD in Sec. 4.5
>
> Thank you for this thoughtful comment. We have substantially revised Section 4.5 to provide a more detailed discussion of the mini-batch SGD setting. Specifically:
>
> - We have restructured the section to clearly explain how our attack works in the mini-batch setting, where adversaries can only access accumulated gradients rather than per-sample gradients
> - We have provided a more comprehensive description of the binning technique, including its role in separating individual sample information from accumulated gradients
> - We have added a step-by-step description of our attack pipeline
>
> > 3. Translation of $\mu$ to $(\varepsilon, \delta)$ in Section 4.1
>
> We sincerely appreciate this suggestion. Conversion of $\mu$ to $(\varepsilon,\delta)$ requires specifying all hyperparameters, specifically the number of steps $T$, sampling probability $p$, and $\delta$. While $\mu$ itself is independent of these parameters, the conversion to $(\varepsilon,\delta)$ is not.
>
> To maintain the broad applicability of our results, we chose not to assume specific values for $T$ and $p$ in the main text. However, we recognize the value of concrete examples to facilitate comparisons with the $(\varepsilon, \delta)$ framework. In response to your suggestion, we have added examples in Appendix C of the revised manuscript. Specifically, we consider typical DP-SGD configurations for CIFAR-10 and ImageNet, as well as a worst-case setting with adversarially chosen hyperparameters. These examples clearly illustrate the substantial impact of training hyperparameters on the resulting privacy guarantees, reinforcing our choice of $\mu$ as the primary metric for our analysis.
>
> > 4. Concurrent Work
>
> We thank the reviewer for bringing these recently released works to our attention. We included a discussion in the related works section of the revised manuscript. We note that their focus and scope differ from ours. These works primarily aim to maximize attack success relative to other reconstruction attacks, whereas our study takes a broader approach. Specifically, we investigate how realistic data priors influence the effectiveness of reconstruction attacks in the context of DP and analyze how these attacks align with or challenge existing theoretical reconstruction bounds. This perspective highlights the practical limitations of current privacy guarantees, providing a complementary rather than overlapping contribution.
>
> > 5. Fully connected input layer
>
> Our focus on fully connected input layers is intentional, as they enable mathematical guarantees for perfect reconstructions under certain conditions (see Eq. 6). This aligns with our goal of analyzing the most powerful attack, providing an upper bound on reconstruction success. Moreover, our threat model allows architectural modifications, making this assumption valid.
>
> While attacks on convolutional layers have been explored, they typically rely on heuristic methods and do not guarantee full reconstruction. We agree that addressing convolutional architectures would broaden the applicability of our attack. In future research, we aim to investigate reconstruction techniques for convolutional layers and compare their performance to fully connected setups.

---

> > ### Comment · Reviewer_j7Cv · 2025-01-21
> >
> > Thank you for your reply. I still have some further comments regarding the response.
> >
> > 1. I still don't think the adversary's ability to extract one image from the batched gradient is a valid setting. This directly violates the setting for DP, specifically, making privacy amplification by subsampling invalid. I tried to obtain the $\sigma$ for training CIFAR-10 using batch size =1, and I cannot get any reasonable $\sigma$.
> >
> > 2. I appreciate the discussion in the App. C on conversion from $\mu$ to $(\epsilon, \delta)$. This also suggests that batch size has to be large enough to match $\mu$ with practical DP training (and adversarial setting).
> >
> > Overall, I don't have other questions.

---

> > > ### Author Response · Authors · 2025-01-22
> > > **Response to Reviewer j7Cv**
> > >
> > > Thank you for your follow-up comments.
> > >
> > > 1. Our experimental results demonstrate that image extraction is feasible without violating DP principles. While privacy amplification by subsampling enhances privacy guarantees, it is an amplification mechanism rather than a fundamental barrier to individual signal extraction. Consider, for instance, a scenario with minimal privacy protection ($\varepsilon=10^9$) - even with privacy amplification, the resulting gradient noise $\sigma$ and the effect on the binning technique for image extraction remain negligible. As privacy protection increases, it more strongly affects the binning technique and reduces extraction quality - an effect our diffusion model-based reconstruction can partially mitigate.
> > >
> > > Regarding your CIFAR-10 experiments with batch size=1, we would be interested in your experimental setup details, particularly whether you used our diffusion model-based reconstruction approach. Our analysis shows that traditional reconstruction methods fail to achieve reasonable $\sigma$ values in this setting.
> > >
> > > 2. Your observation about batch sizes in practical DP training is well-taken. Our analysis deliberately explores edge cases in adversarial scenarios. This approach aligns with traditional DP threat modeling, which assumes maximum adversarial capabilities (knowledge of everything except noise samples and training set membership). While such assumptions may seem extreme compared to typical scenarios, they are crucial for understanding security boundaries. In our work, we analyze a much weaker DP threat model that better reflects real-world scenarios while remaining sufficiently strong to capture important edge cases. As a concrete example, consider a federated learning setting with a malicious central server. If participants don't carefully verify the training process, the server could potentially configure the worst-case settings we analyze (e.g., batch size=1). Even in less obvious attacks (batch size > 1), an adversary might select seemingly performant parameters while optimizing for attack utility. We provide insights into practical vulnerabilities while acknowledging the full spectrum of potential attack scenarios. Please let us know if you would like us to elaborate on any of these points.

---

### Author Response · Authors · 2025-01-15
**Response to Reviewers**

We sincerely thank the reviewers for their thoughtful feedback and positive evaluation of our work. We greatly appreciate the opportunity to clarify and expand on the points raised in the reviews. In response, we have addressed each comment in detail below. Additionally, we have uploaded a revised version of our manuscript, incorporating minor updates following the reviewers' suggestions.

---

### Decision · Action_Editor_f35z · 2025-02-21

**Recommendation:** Accept as is

**Comment:**

As outlined in the "Claims and Evidence" field, the paper studies the privacy properties of computer vision models trained under DP using reconstruction attacks. The paper is well written, and presents the claims and evidence clearly. While one of the three reviewers suggested a weak reject for the paper, I believe the authors have done sufficient job in responding to the criticism during the discussion period, and moreover I believe the paper meets the acceptance criteria of TMLR.

**Audience:**

The paper studies the privacy properties of computer vision model, and therefore would be interesting for the broad audience of TMLR. Furthermore, the empirical findings provide an interesting illustration of the privacy properties of DP training.

**Claims And Evidence:**

In this work, authors argue that using real world image priors as auxiliary information for a reconstruction attack against computer vision models improves the success of the attack. Authors also argue that studying this help to better interpret the privacy protection provided by differential privacy.

The proposed, diffusion model based, attack is applied on three different image data sets. Authors show through various metrics that their attack accuracy lies between an attack performed with no prior information, and an attack that has as a prior a finite selection of images including the target image. Authors further demonstrate how the quality of the reconstructed image is affected by the DP level, providing a nice visual representation of the privacy guarantees.

Reviewers, as well as I, found the set of evidence satisfactory for the claims.